# Nanostructured Electrospun Fibers with Self-Assembled Cyclo-L-Tryptophan-L-Tyrosine Dipeptide as Piezoelectric Materials and Optical Second Harmonic Generators

**DOI:** 10.3390/ma16144993

**Published:** 2023-07-14

**Authors:** Daniela Santos, Rosa M. F. Baptista, Adelino Handa, Bernardo Almeida, Pedro V. Rodrigues, Cidália Castro, Ana Machado, Manuel J. L. F. Rodrigues, Michael Belsley, Etelvina de Matos Gomes

**Affiliations:** 1Laboratory for Materials and Emergent Technologies (LAPMET), Centre of Physics of Minho and Porto Universities (CF-UM-UP), University of Minho, Campus de Gualtar, 4710-057 Braga, Portugal; pg43143@alunos.uminho.pt (D.S.); handa.adelino@gmail.com (A.H.); bernardo@fisica.uminho.pt (B.A.); mrodrigues@fisica.uminho.pt (M.J.L.F.R.); belsley@fisica.uminho.pt (M.B.); emg@fisica.uminho.pt (E.d.M.G.); 2Institute for Polymers and Composites, University of Minho, Campus de Gualtar, 4800-058 Guimarães, Portugal; pedro.rodrigues@dep.uminho.pt (P.V.R.); cidaliacastro@dep.uminho.pt (C.C.)

**Keywords:** cyclo-dipeptide, electrospinning, piezoelectricity, nanofibers, optical second harmonic generation, self-assembly

## Abstract

The potential use of nanostructured dipeptide self-assemblies in materials science for energy harvesting devices is a highly sought-after area of research. Specifically, aromatic cyclo-dipeptides containing tryptophan have garnered attention due to their wide-bandgap semiconductor properties, high mechanical rigidity, photoluminescence, and nonlinear optical behavior. In this study, we present the development of a hybrid system comprising biopolymer electrospun fibers incorporated with the chiral cyclo-dipeptide L-Tryptophan-L-Tyrosine. The resulting nanofibers are wide-bandgap semiconductors (bandgap energy 4.0 eV) consisting of self-assembled nanotubes embedded within a polymer matrix, exhibiting intense blue photoluminescence. Moreover, the cyclo-dipeptide L-Tryptophan-L-Tyrosine incorporated into polycaprolactone nanofibers displays a strong effective second harmonic generation signal of 0.36 pm/V and shows notable piezoelectric properties with a high effective coefficient of 22 pCN${}^{-1}$, a piezoelectric voltage coefficient of $g_{eff}=1.2$ VmN${}^{-1}$ and a peak power density delivered by the nanofiber mat of 0.16μWcm${}^{-2}$. These hybrid systems hold great promise for applications in the field of nanoenergy harvesting and nanophotonics.

## 1. Introduction

Nanotechnology has significantly impacted various industries, including bionanotechnology. It has advanced medical imaging, targeted drug delivery, customized medical devices, and disease-specific therapeutics [1]. In energy generation and storage, nanotechnology has improved efficiency and sustainability [2]. Additionally, bionanotechnology has led to the creation of novel materials and sensors for the detection and removal of environmental pollutants [3].

Peptides, consisting of short amino acid chains, exhibit diverse structures, functions, and biological activities [4]. Among them, cyclic dipeptides or cyclo-dipeptides (CDPs) are notable. CDPs form ring structures through peptide bonds and possess enhanced bioactivity, structural rigidity, biofunctionalization, and enzymatic stability. These unique characteristics make CDPs promising for various bionanotechnology applications [5,6].

CDPs exhibit antimicrobial and anticancer properties. They can self-assemble into nanostructures like nanotubes, nanofibers, nanowires, and nanospheres, offering unique properties based on their sequence and environment [7].

Through engineering, CDPs self-assemble into nanoscale structures such as nanotubes, nanofibers, nanowires, and nanospheres with unique properties [8,9]. Aromatic groups in organic materials, including dipeptide nanostructures, exhibit enhanced rigidity and mechanical, optical, and piezoelectric properties when self-assembled [10,11].

Self-assembly occurs through non-covalent interactions such as hydrogen bonding, electrostatic interactions, Van der Waals forces, π-π and hydrophobic interactions [12]. Combining CDPs with other nanomaterials and technologies enables the creation of novel materials and devices applicable in tissue engineering, drug delivery, biosensors, and diagnostics [6,8,9,13]. CDPs, particularly those containing tryptophan, self-assemble through supramolecular phases involving hydrogen bonding and aromatic interactions, resulting in improved conductivity, photoluminescence, mechanical strength, and piezoelectric properties [8,13,14,15]. Incorporating cyclic dipeptides into polymer fibers enhances mechanical properties, making them suitable for composite materials, the manufacturing of microelectronic devices, and tissue engineering [16]. Electrospinning, a technique utilizing electrostatic forces, can be employed to create fibers incorporating CDPs within a polymeric matrix. Factors such as polymer solution properties, flow rate, electric field strength, needle-to-collector distance, and environmental conditions influence the electrospinning process, affecting fiber morphology and diameter. Controlling these factors is crucial to optimizing nanofiber performance [17,18].

Studying second harmonic generation and piezoelectric properties in hybrid systems with cyclic dipeptides is vital for creating advanced materials and devices in various fields. Second harmonic generation is a nonlinear optical property that allows for the efficient conversion of a single light beam into two photons of higher energy [19,20,21]. Piezoelectricity, on the other hand, involves the generation of electric charge under mechanical stress and is essential in pressure sensors, mechanical energy devices, and energy conversion systems [22]. Exploring and understanding these properties in cyclic dipeptide-incorporated fiber systems can lead to the development of innovative materials.

This study represents an innovative contribution in the field, as it is the first time that Cyclo-L-Tryptophan-L-Tyrosine dipeptides have been integrated into a fibrous matrix. We used the electrospinning technique to promote the immediate self-assembling of nanostructures within the fibers. As a result, we achieved a nanofibrous structure with piezoelectric and optical second harmonic generation properties. It is important to emphasize that self-assembling is typically reported in solution, making this approach particularly innovative.

## 2. Materials and Methods

### 2.1. Materials

The chiral cyclic dipeptide Cyclo-L-Tryptophan-L-Tyrosine, hereafter referred to as Cyclo(L-Trp-L-Tyr) (Mw = 349.39 g/mol), was purchased from Bachem AG (Bubendorf, Switzerland), and its chemical formula is represented in Figure 1.

The biopolymer chosen was polycaprolactone (PCL, Mw = 80,000 g/mol) purchased from Sigma-Aldrich (Darmstadt, Germany). Its chemical formula is also illustrated in Figure 1.

Regarding the solvents used, 1,1,1,3,3,3-hexafluoro-2-propanol (HFP), dichloromethane (DCM), *N,N*-dimethylformamide (DMF), *N,N*-dimethylacetamide (DMAc) and methanol (MeOH), they were purchased from Merck/Sigma-Aldrich (Darmstadt, Germany). All chemicals were used as received.

### 2.2. Dipeptide Self-Assembled Nanostructures in Solution

To visualize the nanostructures to which the dipeptide self-assembles, a solution was prepared by dissolving 0.80 mg of Cyclo(L-Trp-L-Tyr) in the following mixture of solvents: 4:2:1 DMF/DMAc/H${}_{2}$O. This solution was filtered with a membrane with a 0.45 μm pore diameter and allowed to grow the nanostructures for a month. A drop of this solution was then placed on a silica slide, and the solvents were slowly evaporated at room temperature and sent for scanning electron microscopy (SEM) analysis.

To study the optical absorption and photoluminescence spectra, the dipeptide was dissolved in HFP at a concentration of 100 mg/mL. Afterwards, it was diluted in MeOH to the final concentrations depending on the characterization and left at room temperature for 24 h.

### 2.3. Electrospinning of Nanofibers

For electrospinning, a 10% polymer solution of PCL (*w/v*) was completely dissolved in DCM at 300 rpm and 35 °C. A mixture of solvents DMF/DMAc was used to dissolve Cyclo(L-Trp-L-Tyr) in a 1:5 *w/w*. This was incorporated into the polymer solution, resulting in a 4:1:1 volume ratio of solvents. At room temperature and for several hours, the obtained solution was stirred at 300 rpm.

The vertical electrospinning used, under ambient conditions, has a high voltage power supply with an electrical potential difference of 18 kV, from Spellmann CZE2000, Bochum, Germany. The tip-of-needle collector distance (TCD) was 11 cm, and a high purity aluminum foil was placed on the collector to facilitate fiber manipulation and act as an electrode.

Other electrospinning parameters to ensure a stable process and the production of bead-free nanofibers were the flow rate of 0.18 mL/h and a needle with an outer diameter of 0.813 mm (inner diameter 0.508 mm). The syringe was loaded with the polymeric and dipeptide solution and the high voltage power of 18 kV was applied to the tip of needle, as seen in Figure 1.

### 2.4. Scanning Electron Microscopy (SEM)

The morphological characterization of the dipeptide was investigated using SEM. This analytical technique not only enables the identification of the nanostructures that the CDP self-assembles into, but also allows for the examination of their size distribution. The primary objective of this study is to investigate and compare the nanostructures formed in solution with those formed during the process of electrospinning within the polymeric matrix.

To study the morphology of Cyclo(L-Trp-L-Tyr) structures, scanning electron microscopy was performed using a Nova Nano SEM 200 microscope, operated at an accelerating voltage of 10 kV. Slides with Cyclo(L-Trp-L-Tyr) structures in solution and the fiber mat were covered with a 10 nm thick Au-Pd (gold-palladium) film (80–20 weight%) using a high resolution sputter coater, 208HR Cressington Company, coupled to a high resolution thickness controller, MTM-20 Cressigton. The resulting images were analyzed using ImageJ 1.53 k image analysis software (NIH, https://imagej.nih.gov/ij/, 13 February 2023). The distribution and average diameters were calculated by measuring 145 nanofibrils and 88 fibers from the SEM images, and the results were fitted to a log-normal function.

### 2.5. Confocal Laser Scanning Microscopy

By employing this photoluminescent technique, it becomes possible to observe the distribution of the active material (CDP) in the interior of the nanofibers.

The Olympus™ FluoView FV1000 confocal scanning laser microscope (Olympus, Tokyo, Japan) was used to observe the fluorescence of the nanofibers with a 40× objective, an excitation wavelength of 405 nm and detection filters BA 430–470. Confocal images were acquired with a 800 × 800 pixel resolution and a 1 cm${}^{2}$ fiber mat with a 120 μm thickness on a glass slide. The sample was scanned at room temperature.

### 2.6. Optical Absorption, Photoluminescence and Diffuse Reflectance Spectroscopy (DRS)

The optical characterization of the fibers using techniques such as optical absorption, photoluminescence, and diffuse reflectance spectroscopy is crucial for understanding their electronic and optical properties. These techniques enable the determination of the bandgap, which is essential for evaluating the semiconducting properties of the fibers and their potential in optical and electronic applications.

Optical absorption (OA) measurements were performed on Cyclo(L-Trp-L-Tyr) solutions and fiber mat using a Shimadzu UV-3600 Plus UV–Vis–NIR (ultraviolet/visible/near-infrared) spectrophotometer (Shimadzu Corporation, Kyoto, Japan). At room temperature and in the wavelength range of 200 to 600 nm, the solutions were prepared in methanol and measured in a quartz cuvette with a path length of 1 cm.

From UV–vis absorption spectroscopy, it is possible to determine the optical bandgap energy $\left(E_{g}\right)$ of the nanoparticles using the Tauc plot given by ${\left(\alpha h\nu \right)}^{n}=k\left(h\nu -E_{g}\right)$[23]. The parameters are the following: α is the absorption coefficient; hν is the incident photon energy; n, is the type of electronic transition and, for this work, n=1/2, as this is an indirect bandgap material; *k* is a constant of proportionality independent of the energy and k=1 (amorphous materials).

The equipment used for the photoluminescence (PL) spectra was a Fluorolog 3 spectrofluorimeter (HORIBA Jobin Yvon IBH Ltd., Glasgow, UK) using an excitation wavelength of 280 nm and in the wavelength range of 290 to 600 nm, with input and output slits fixed to provide a spectral resolution of 2 nm. Photoluminescence excitation (PLE) spectra were recorded in the wavelength range of 220 to 300 nm.

For the measurement of the diffuse reflectance spectrum (DRS) of the nanofiber mat, a UV-2501PC spectrophotometer (Shimadzu Corporation, Kyoto, Japan) equipped with an integration sphere, Shimadzu ISR-205 240A, was used, and barium sulfate was taken as a reference. In the measurement, a wavelength rate of 200 to 800 nm was used with a step size of 1 nm.

To determine the bandgap energy $\left(E_{g}\right)$ of the material, we applied the Kulbelka–Munk function, F(R), to the experimental data obtained from DRS analysis [24]. The Kubelka–Munk function is given by $F\left(R\right)={\left(1-R\right)}^{2}/2R$, where R is the total reflectance coefficient of the material. As such, the bandgap is calculated using equation ${\left[h\nu F\left(R\right)\right]}^{1/2}=\alpha \left(h\nu -E_{g}\right)$ [25].

### 2.7. Mechanical and Hydrophobic Tests

The mechanical and hydrophobic characterization of the fibers is essential for their application as piezoelectric nanogenerators. Through the measurement of mechanical properties such as tensile strength and elastic modulus, we can determine their capacity to generate piezoelectric energy under mechanical stress, which is crucial for the operation of the nanogenerators. Additionally, the characterization of fiber hydrophobicity through contact angle measurements is relevant for applications in humid environments, ensuring the efficiency and stability of piezoelectric nanogenerators.

Mechanical tests were performed on a Zwick/Roell Z005 (ZwickRoell, Ulm, Germany) universal testing machine (ASTM D882-02). The specimens were cut from electrospun PCL fiber mats (40 × 10 mm) and tested at 25 mm/min with a gauge length of 26 mm. The mechanical indexes are the average values of at least 7 specimens.

The hydrophilicity of the electrospun PCL fiber mat was measured using a goniometer (Contact Angle System OCA 20 Dataphysics, Filderstadt, Germany). Deionized water (3 μL) was automatically dropped onto the flat fibers with a precision syringe, following the sessile drop method. At least 20 measurements were made and the mean value was taken.

### 2.8. Second Harmonic Generation (SHG)

To enhance the characterization of the optical properties of the fibrous material, analyze the crystalline structures of the nanostructures within the fibers, and evaluate nonlinear processes, comprehensive studies of second harmonic generation were conducted.

Second harmonic measurements on the fiber mats were performed using a mode-locked Ti: Sapphire laser (Coherent Mira) as an excitation source. The incident fundamental light was focused on the fiber mats using a Nikon CFI Plan Fluor ×10 objective; see Figure 2. Due to the opacity of the fiber mats, the second harmonic light was detected in reflection (epi-illumination). Although the bandwidth of the fundamental light is sufficient to produce 85 fs pulses (full-width half-maximum), we estimate that dispersion in the optical elements stretches the pulses to an approximately 120 fs duration when incident on the samples. For the fibers, incident pulse energies ranged from 65 to 130 pJ. A 1 mm thick beta barium borate (BBO) crystal cut for phase matching at 800 nm was used to calibrate the detection system. For the BBO crystal, the incident pulse energies were 5 to 7 pJ. The second harmonic light was separated from the incident fundamental light by a short-pass dichroic mirror; then, it passed through a narrow band-pass filter (Semrock FF01-405/150-25) before being focused onto a fiber bundle coupled to an Andor imaging spectrometer (Shamrock 300i) equipped with a charge-coupled device, CCD, array (Newton) cooled to −50 °C. The signal was integrated for 1 s in the case of the fibers and 4 ms in the case of BBO. The second harmonic spectra were fit to a Gaussian profile as detailed in the supplementary information of [26]. The area under the Gaussian fit was taken as a proxy for the total number of second harmonic signal counts.

### 2.9. Piezoelectric Measurements

The measurements of the piezoelectric properties were conducted using a well-defined experimental setup (see Appendix A). This setup ensured accurate and reliable measurements, which are essential for validating and reproducing the results obtained for the piezoelectric properties of the fibers.

The piezoelectric output voltage was measured through a load resistance of 100 MΩ and collected in a digital storage oscilloscope (Agilent Technologies DS0-X-3012A3012A, Waldbronn, Germany) after going through a low-pass filter and a low noise preamplifier (Research systems SR560, Stanford Research Systems, Stanford, CA, USA). The nanofiber sample had a 30 × 40 mm${}^{2}$ area and a thickness of 120 μm and was placed in thin plates of high-purity copper (top: 23 × 30 mm${}^{2}$ and bottom: 30 × 33 mm${}^{2}$). Periodic mechanical forces were applied to the fiber array by a vibration generator (Frederiksen SF2185) with a frequency of 3 Hz determined by a signal generator (Hewlett Packard 33120A). Before measuring, it was necessary to calibrate the applied forces with a FSR402 force-sensing resistor (Interlink Electronics Sensor Technology, Graefelfing, Germany). The forces applied were uniform and perpendicular to the surface area of the sample.

## 3. Results and Discussion

### 3.1. Morphological Characterization

The dipeptide Cyclo(L-Trp-L-Tyr) in a solution of DMF/DMAc/H${}_{2}$O self-assembles into nanofibrils, as may be seen in the SEM images in Figure 3a,b. These nanofibrils have an average diameter of 218 ± 70 nm as shown in Figure 3c. The histogram was constructed from a total of 145 measurements, with the minimum diameter of 87 nm and the maximum diameter of 487 nm.

Figure 3d,e are the SEM images of the dipeptide embedded into nanofibers, Cyclo(L-Trp-L-Tyr)@PCL, fabricated by the electrospinning technique. These fibers have an average diameter of 417 ± 112 nm and a quite wide diameter distribution from a minimum diameter of 208 nm to a maximum of 696 nm, Figure 3f. In Figure 3e, it is possible to see nanotubes formed outside the fiber, with an average diameter of 58 ± 13 nm, indicating that in the polymer matrix, the dipeptide self-assembles in the form of nanotubes.

In hydrogel form, You Y. and co-authors successfully produced Cyclo(L-Trp-L-Tyr) dipeptide fibrils measuring 100 ± 50 nm in width. Furthermore, they observed interconnected fibrous networks in 3D, formed by these fibrils [27].

In our previous work [16], we verified that the dipeptide Cyclo(L-Trp-L-Trp) self-assembles into nanospheres with an average diameter of 245 nm.

In the current study, we utilized the dipeptide Cyclo(L-Trp-L-Tyr), taking advantage of the tyrosine OH group’s ability to form additional hydrogen bonds with both the tryptophan units and within the tyrosine residues. These hydrogen bonds, facilitated by the tyrosine OH group, are considered stronger than those formed by the tryptophan NH group. This increased strength is attributed to the higher electronegativity of oxygen in the tyrosine OH group compared to nitrogen in the tryptophan NH group, resulting in a greater attraction of shared electrons within the hydrogen bond [27,28]. Furthermore, the linear arrangement of the hydroxyl (OH) group in tyrosine contributes to enhanced interactions. This is due to the electron-withdrawing nature of the hydroxyl substitution on the phenyl side chain of tyrosine, promoting more efficient π-π interactions involving the side chains [29]. Additionally, it is important to note that tryptophan exhibits a higher hydrophobicity index (1.9) compared to tyrosine (−0.7) [30]. These differences in hydrophobicity may play a role in the self-assembling process. When combined with the enhanced strength of the hydrogen bonds, they facilitate the formation of a more extensive and organized network, thus promoting the formation of nanotubes.

In terms of electrospun nanofibers, they showed no bead formation, and electrospinning proved to be a stable and appropriate technique for the production of these nanomaterials.

In article [16], we obtained an average diameter of 600 nm for PCL nanofibers functionalized with Cyclo(L-Trp-L-Trp), and in [31], the electrospun fibers of Boc-Phe-Phe (Boc-L-phenylalanine-L-phenylalanine-OH) formed from PCL resulted in diameters in a range of 590–700 nm, similar to the results from the present study.

An additional structural characterization was performed by Fourier transform infrared spectroscopy (FTIR); see Appendix A. The spectra show changes in peak intensity and shape, indicating the successful incorporation of the CDP inside the fibers.

### 3.2. Optical Absorption, Photoluminescence and Diffuse Reflectance Spectroscopy of Dipeptide Self-Assemblies

The UV–vis absorption spectrum of a 0.43 mM Cyclo(L-Trp-L-Tyr) in methanol is depicted in Figure 4a, showing an absorption band from 250 to 300 nm with three smaller spikes at 273, 280 and 289 nm. These represent the formation of structures called quantum dots, QDs. As reported in the literature, for the case of Cyclo-FW (cyclo-phenylalanine–tryptophan) and Cyclo-WW (cyclo-tryptophan–tryptophan), the monomers of CDPs form dimeric QDs that act as the foundation for more complex supramolecular structures [8].

The bandgap energy of $E_{g}=4.11\pm 0.01$ eV was obtained for Cyclo(L-Trp-L-Tyr) from the absorption spectrum through the Tauc plot; see the inset in Figure 4a. The obtained value agrees with those reported from density functional theory (DFT) for Cyclo-FW and Cyclo-WW, which have bandgap energies of 3.63 eV and 3.56 eV, respectively [8].

In Figure 4b, it is possible to verify the match between the absorption spectrum and the photoluminescence excitation (PLE) spectrum of the dipeptide in solution for excitation at a maximum wavelength of 280 nm.

The photoluminescence (PL) emission spectrum of the 0.001 mM dipeptide solution in MeOH for excitation at 280 nm is represented in Figure 4b with maximum emission peak at 311 nm. The PL spectrum of Cyclo(L-Trp-L-Tyr)@PCL nanofibers is represented in blue, also in Figure 4b, with a maximum at 330 nm and another, less intense emission peak at 440 nm, in the blue region of the optical spectra. That might indicate the presence of larger dipeptide nanostructures. The shift observed in the PL spectrum, from 311 nm for a dipeptide solution to 330 nm when the dipeptide is embedded into the nanofiber, is representative of an increase in the size of the nanostructures incorporated into the nanofibers.

As a further study of the nanofibers, the diffuse reflectance spectrum (DRS) was measured for Cyclo(L-Trp-L-Tyr)@PCL and compared with the fiber mat without the embedded dipeptide in Figure 4c. The spectrum shows two peaks of minimum reflectance at 283 and 330 nm. The bandgap energies obtained for the dipeptide nanostructures both in solution and embedded into the electrospun fibers are approximately 4.0 eV, which are expected values for organic wide-bandgap semiconductors [32].

The bandgap energy values obtained in this study for Cyclo(L-Trp-L-Tyr) are also similar to those reported in a previous work with Cyclo(L-Trp-L-Trp) in solution and in fibers [16].

In Figure 4d, the confocal microscopy image of Cyclo(L-Trp-L-Tyr)@PCL nanofibers is presenting, showing intense blue photoluminescence due to CDP nanostructures in the interior of the fibers.

### 3.3. Mechanical and Hydrophobic Properties of Electrospun Fibers

In Figure 5, the mechanical properties of PCL fibers are presented, with stress–strain curves indicating that the incorporation of CDPs increases stiffness. On average, the elastic modulus (Young’s modulus) increased by 445% (from 6.9 to 37.6 MPa) and the tensile strength increased by 72% (from 1.5 to 2.6 MPa), while the elongation at break was reduced by 35% (from 121 to 79%). Additionally, in a previous study, a notable improvement in the mechanical characteristics of the polymer was observed with the integration of organic crystals, specifically involving polyvinyl chloride (PVC) [33].

The cyclo-dipeptides effectively transfer applied forces between the polymer and nanotubes, reinforcing the polymeric matrix. This behavior is similar to that observed for electrospun polymer fibers embedded with nanospheres of the Cyclo(L-Trp-L-Trp) dipeptide [16] where the fiber mechanical properties were improved. These results, an increase in the elastic modulus and an elongation at break, are beneficial to the piezoelectric response of the material, enabling the fiber mat to withstand greater forces and deformations. Young’s modulus was found to depend significantly on the angle between the stretch direction and the fiber direction [34]; in our case, the functional electrospun fibers were stretched along the longitudinal axis of the fiber. Our previous work also demonstrates that the incorporation of lead-free organic ferroelectric perovskite N-methyl-N’-diazabicyclo [2.2.2]octonium-ammonium triiodide (MDABCO-NH${}_{4}$I${}_{3}$) nanocrystals in polyvinyl chloride microfibers [33] resulted in an improvement in Young’s modulus compared to polymer electrospun fibers with no embedded material, as shown in Table 1.

The hydrophilic/hydrophobic characteristics of biodegradable polymers can be determined by their absorption of water, which, in turn, affects their susceptibility to degradation by hydrolytic processes. It is known that PCL is a hydrophobic polymer [37,38].

Figure 6 shows the contact angles of PCL fibers and Cyclo(L-Trp-L-Tyr)@PCL fibers, and it is possible to notice a significant reduction in the hydrophobicity of PCL fibers after the incorporation of Cyclo(L-Trp-L-Tyr) nanotubes. This reduction is due to the presence of a large number of hydroxyl groups (-OH) and a wide network of hydrogen bonds established in the peptide nanostructures that strongly bind to water molecules. When water molecules are adsorbed by the peptide nanostructures of Cyclo(L-Trp-L-Tyr), they can form highly mobile hydrogen ions that facilitate the transfer of electrical charge through the material. This mobility of hydrogen ions might increase the electrical conductivity of the material in response to higher levels of humidity.

Furthermore, the thermal behavior of Cyclo(L-Trp-L-Tyr) dipeptide, Cyclo(L-Trp-L-Tyr)@PCL and PCL nanofibers was tested by thermogravimetric analysis (see Appendix A) and differential scanning calorimetry (Appendix A). The findings presented herein suggest that the introduction of the dipeptide had an insignificant effect on the thermal properties of the biopolymer. Notably, the thermal stability of the polymer matrix enables it to withstand temperatures of up to 405 ºC without undergoing degradation. This remarkable attribute makes it highly suitable for the envisaged technological applications.

### 3.4. Second Harmonic Generation of Nanofibers

To estimate the effective second-order nonlinear susceptibility, $d_{eff}$, of the Cyclo(L-Trp-L-Tyr) nanocrystals embedded in the PCL fibers, the second harmonic light measured in reflection from the fibers was compared to that generated by a 1 mm thick orientated beta barium borate (BBO) crystal.

For the calibration signal, a 10× microscope objective (CFI Plan Fluor from Nikon) was used to focus the 4 mm diameter incident fundamental beam within the BBO crystal. Given the effective focal length of 20 mm, we estimate the beam waist at focus is approximately 2.5 μm, leading to a Rayleigh length (in air) of $z_{R}=25\;\;\mu$m. Under conditions of strong spatial walk-off due to birefringence, the heuristic model of Wang and Wiener [39] predicts that the second harmonic generation efficiency for an ultrafast pulse will obey the following equation:$$\eta_{2\omega }^{BBO}\equiv \frac{U_{2\omega }^{BBO}}{U_{\omega }^{BBO}}=\frac{{\left[2\omega d_{eff}^{BBO}\right]}^{2}}{n_{\omega }n_{2\omega }\lambda_{0}c^{3}\varepsilon_{0}}\;\pi \;\sqrt{\frac{\ln2}{2}}\;l_{S}^{BBO}\;\frac{U_{\omega }^{BBO}}{t_{p}}.$$

Here, $U_{2\omega }$ and $U_{\omega }$ are, respectively, the energies of the generated second harmonic and incident fundamental pulses, $t_{p}$ is the full-width half-maximum incident pulse duration (which we estimate to be approximately 120 fs). At the phase matching angle, the refractive indices of the fundamental and second harmonic light are $n_{\omega }=n_{2\omega }=1.66$, while for incident fundamental light at 800 nm, $d_{eff}^{BBO}=2.0$ pm/V [40]. The parameter $l_{S}^{BBO}$ is an effective crystal thickness given by the ratio of the focused beam waist to the walk-off angle of the second harmonic light ρ=69 mrad that propagates as an extraordinary wave. We estimate that $l_{S}^{BBO}≃36\;\;\mu$m.

On the other hand, since the Cyclo(L-Trp-L-Tyr)@PCL fibers have a thickness of the order of the fundamental wavelength, one can invoke the standard plane-wave result for the second harmonic generation leading to an efficiency given by
$$\eta_{2\omega }^{Cyclo(L-Trp-L-Tyr)@PCL}\equiv \frac{{\left[2\omega \;d_{eff}^{Cyclo(L-Trp-L-Tyr)@PCL}\right]}^{2}}{n_{\omega }n_{2\omega }\lambda_{0}c^{3}\varepsilon_{0}}\;\sqrt{\frac{\ln2}{2\pi }}\;\frac{U_{\omega }^{Cyclo(L-Trp-L-Tyr)@PCL}s_{Cyclo(L-Trp-L-Tyr)@PCL}^{2}}{z_{R}\;t_{p}}.$$

Here, $s_{Cyclo(L-Trp-L-Tyr)@PCL}$ is the thickness of the illuminated Cyclo(L-Trp-L-Tyr) crystal embedded in the PCL fibers. Taking the ratio between these two efficiencies, we can solve the effective nonlinear susceptibility of the Cyclo(L-Trp-L-Tyr) nanocrystals,
$$d_{eff}^{Cyclo(L-Trp-L-Tyr)@PCL}≃d_{eff}^{BBO}\left(\frac{U_{\omega }^{BBO}}{U_{\omega }^{Cyclo(L-Trp-L-Tyr)@PCL}}\right)\sqrt{\frac{U_{2\omega }^{Cyclo(L-Trp-L-Tyr)@PCL}}{U_{2\omega }^{BBO}}}\;\pi^{3/4}\frac{\sqrt{z_{R}\;l_{SWO}^{BBO}}}{s_{Cyclo(L-Trp-L-Tyr)@PCL}}.$$

We simplified the above by neglecting small differences between the refractive indices of the BBO and the Cyclo(L-Trp-L-Tyr) nanocrystals. Finally, due to the opaqueness of the fiber mats, we measured the second harmonic generation from the fiber mats in the reflection. Since the above expression is for transmission, it was also necessary to calibrate the relative efficiency between the transmission and reflection geometries. Using the BBO crystal, we found that the second harmonic signal measured in transmission was approximately 47 times larger than that in reflection.

Rather than measuring directing the second harmonic signal energies, we used the amplitude of a Gaussian fit to the acquired second harmonic spectra as a proxy. An example of a fit to the second harmonic light generated by the fiber mat is shown below in Figure 7, where the small and random nature of the residuals is evident.

The measurements carried out on BBO in transmission were taken with an average incident energy of 5.3 pJ per pulse with a pulse repetition rate of 76 MHz. The second harmonic spectra were integrated over 4 ms and the Gaussian fits to the spectra provided an average count rate at the spectral peak of 8.3 × 10${}^{6}$ counts/s. For the Cyclo(L-Trp-L-Tyr)@PCL fiber mat, a higher incident energy of 129 pJ was obtained. The second harmonic spectra were integrated over 1 s and the count rate at the spectral peak was measured to be approximately 102 counts/s. We note, however, that a single fiber has an average diameter of 417 nm, much smaller than the incident fundamental beam waist of 2.5 mm. We found that significant second harmonic generation from the fiber mats was observed only at select positions, which leads us to hypothesize that only a few fibers contain Cyclo(L-Trp-L-Tyr) nanocrystals with the correct orientation to generate significant second harmonic light. Assuming that the second harmonic signal from the fiber mats is dominated by a single fiber illuminated in the beam waist, the effective incident energy should be reduced by the ratio of the fiber cross-section to the transverse profile, weighted by the Gaussian spatial profile of the incident beam which we estimate to be roughly 0.13. Putting all the measurements together, we estimate that the effective second-order nonlinear susceptibility of the Cyclo(L-Trp-L-Tyr) nanocrystals is
$$d_{eff}^{Cyclo(L-Trp-L-Tyr)@PCL}≃0.36\pm 0.13\;\;\mathrm{pm}/V.$$

In this estimate, the uncertainty arises solely from the estimated standard deviation of the fiber diameters (112 nm).

This result is approximately three times greater than that of the effective second-order nonlinear susceptibility value of the dipeptide glycyl-L-alanine hydroiodide monohydrate (Gly-L-Ala.HI.H${}_{2}$O) crystals [26].

### 3.5. Piezoelectric Response of Cyclo(L-Trp-L-Tyr) Nanofibers

Due to its chirality, crystalline Cyclo(L-Trp-L-Tyr) is piezoelectric. In the functionalized fibers, piezoelectricity arises mainly from the dipeptide, as it is known that the biopolymer PCL is not piezoelectric.

The voltage and current obtained from a Cyclo(L-Trp-L-Tyr)@PCL fiber mat was measured when perpendicular periodic forces were applied to the sample due to the transformation of mechanical energy into an electrical response, as shown in Figure 8b (see the inset). The application of periodic forces on the fiber mat causes a reorientation of the molecular dipoles inside the crystalline material accompanied by a charge separation which originates an output voltage across the active material, and finally an electric current is generated through the external circuit.

Figure 8 shows that electrospun PCL nanofibers, when funcionalized with Cyclo(L-Trp-L-Tyr), behave as a piezoelectric nanogenerator (PENG).

The maximum value of the measured output voltage was 10.5 V and the maximum piezoelectric current was 105 nA for a maximum applied force of 4.8 N; see Figure 8a. Figure 8b shows the linear proportionality between the piezoelectric current and the applied periodical forces.

To determine the piezoelectric coefficient, the following steps were followed: the piezoelectric charge (Q) was calculated from the piezoelectric current I, $Q=\int_{}^{}Idt\;\;\left(C\right)$, considering a response time of 0.001 s; defining $d_{eff}=Q/F$ (pCN${}^{-1}$), the effective piezoelectric coefficient for the Cyclo(L-Trp-L-Tyr)@PCL nanofibers was 22 pCN${}^{-1}$. In Table 2, this quantity, obtained from the measured electric current, is presented for several dipeptides for comparison. It is also mentioned in Table 2 that for an electrospun fiber mat fabricated with lead-free organic–inorganic perovskite N-methyl-N’-diazabicyclo [2.2.2]octonium-ammonium triiodide (MDABCO-NH${}_{4}$I${}_{3}$) with PVC, $d_{eff}=175$ pCN${}^{-1}$ [33].

In a previous study [16], for the dipeptide Cyclo(L-Trp-L-Trp) incorporated in PCL fibers, the piezoelectric coefficient obtained was 30 pCN${}^{-1}$ (Table 2), slightly higher than the value obtained in this work but within the same order of magnitude.

It is interesting to compare the effective piezoelectric coefficient obtained for Cyclo(L-Trp-L-Tyr)@PCL with that of Boc (N-*tert*-butoxycarbonyl)-protected linear dipeptide, Boc-L-phenylalanyl-L-tyrosine, embedded in the poly(L-lactic acid) (PLLA) fibers, which is around 7 pCN${}^{-1}$ [41], three times smaller. The results presented in Table 2 allow the conclusion that cyclic dipeptides embedded into electrospun fibers display higher piezoelectric properties than the correspondent dipeptides under the form of polycrystalline powder; it is also more advantageous for linear dipeptides to be used in energy harvesting devices through the piezoelectric effect.

Aiming at quantifying the nanofiber mats as a piezoelectric nanogenerator, a figure of merit given by the piezoelectric voltage coefficient $g_{eff}=d_{eff}/\left(\varepsilon^{\prime }\varepsilon_{0}\right)$ VmN${}^{-1}$ needs to be calculated. From the reported data [42], the dielectric constant of Cyclo(L-Trp-L-Tyr)@PCL is $\varepsilon^{\prime }∼2$ at low frequencies. Therefore, we obtain $g_{eff}=1.2$ VmN${}^{-1}$ for the Cyclo (L-Trp-L-Tyr)@PCL fiber mat. This value is within the same order of magnitude as that calculated for the Cyclo (L-Trp-L-Trp)@PCL fiber mat $(g_{eff}=2.6$ VmN${}^{-1})$.

Finally, the peak power density given by $P=\left(RI^{2}\right)/A\;(\mu$Wcm${}^{-2})$, where R=100MΩ is the load resistance and A is the electrodes area, delivered by the nanofiber mat is 0.16 μWcm${}^{-2}$, very similar to the power density delivered by Cyclo (L-Trp-L-Trp)@PCL fibers 0.13 (μWcm${}^{-2})$

From Table 2, we arrive at the conclusion that cyclic dipeptides embedded into electrospun fibers display higher piezoelectric properties than those of polycrystalline powder of cyclic dipeptides and linear dipeptides. Therefore, they show great potential to be used in energy harvesting devices through the piezoelectric effect.

## 4. Conclusions

Biopolymer polycaprolactone (PCL) electrospun nanofibers with self-assembled cyclic dipeptide L-Tryptophan-L-Tyrosine nanotubes (Cyclo(L-Trp-L-Tyr)@PCL) form a flexible and mechanically resistant fiber mat, showing intense blue photoluminescence. The dipeptide nanotubes are wide-bandgap semiconductors with a bandgap energy of 4.0 eV as calculated from the diffuse reflectance spectrum measured in the range of 300 nm to 800 nm. When embedding the dipeptide into the polymer matrix, there is an increase in the Young’s modulus by 445% and the tensile strength by 72% when compared with dipeptide free polymer fibers.

Additionally, the Cyclo(L-Trp-L-Tyr)@PCL fibers demonstrate great potential as optical second harmonic nanogenerators as they present a nonlinear optical effective coefficient of 0.36 pm/V. This value is approximately six times lower than the corresponding coefficient observed in an optical phase-matched inorganic beta barium borate single crystal. Although the effective coefficient is smaller, for nanophotonic applications, it is much simpler and more manageable to work with dipeptide polymer nanofiber mats than with inorganic brittle single crystals that have to be cut with a specific crystallographic orientation.

In order to investigate the feasibility of using this chiral cyclo-dipeptide as a piezoelectric energy harvesting device, periodic forces were applied to the fiber mat, revealing a strong effective piezoelectric coefficient of 22 pCN${}^{-1}$. Furthermore, the piezoelectric voltage coefficient is $g_{eff}=1.2$ VmN${}^{-1}$ and the peak power density delivered by the nanofiber mat is 0.16 μWcm${}^{-2}$, very similar to those reported for Cyclo(L-Trp-L-Trp)@PCL fibers, which are, respectively, 2.6 VmN${}^{-1}$ and 0.13 μWcm${}^{-2}$.

As a final conclusion, the obtained results provide evidence for the considerable potential of the cyclic-dipeptide L-Tryptophan-L-Tyrosine when incorporated into electrospun PCL nanofibers as optical devices and as nanogenerators for energy harvesting technologies.

## Figures and Tables

**Figure 1 materials-16-04993-f001:**
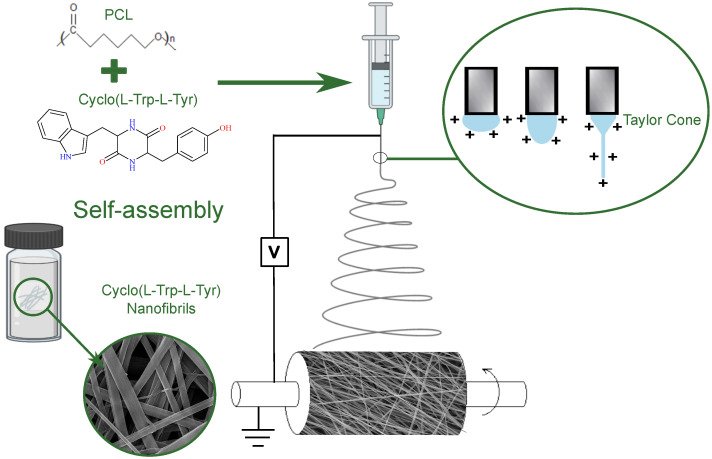
Schematic representation of the electrospinning process and the materials used, both in the production of the Cyclo(L-Trp-L-Tyr)@PCL nanofibers and the self-assembly in solution of dipeptide.

**Figure 2 materials-16-04993-f002:**
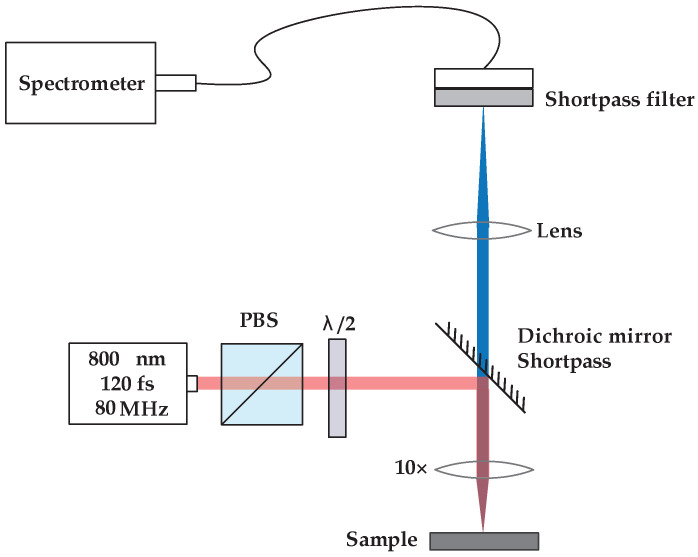
Second harmonic microscope layout. PBS—Polarized beam splitter; λ/2—half-waveplate. The transmission of the PBS is aligned vertically.

**Figure 3 materials-16-04993-f003:**
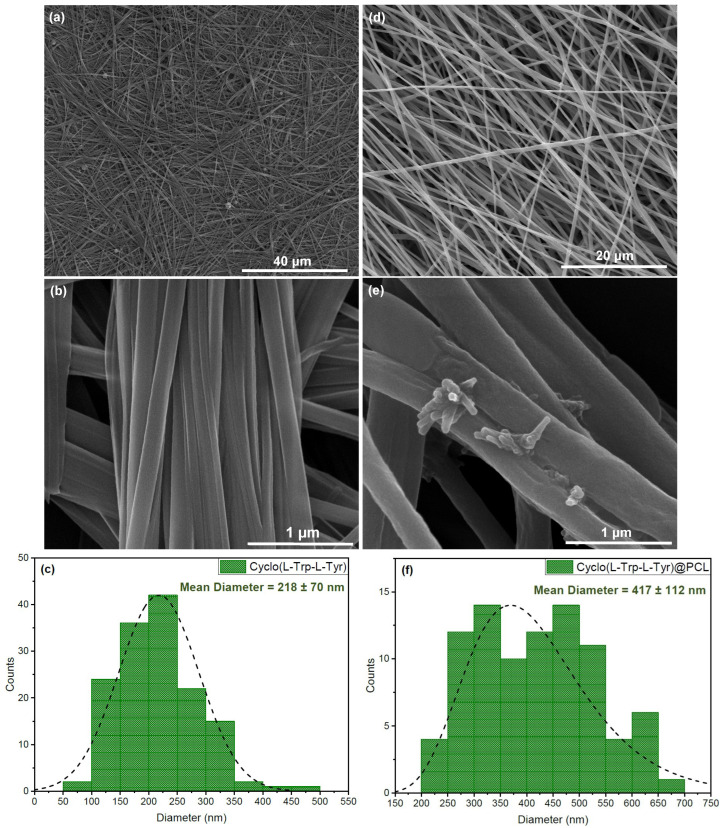
SEM images of Cyclo(L-Trp-L-Tyr): (**a**,**b**) as self-assembled nanofibrils in a solution of DMF/DMAc/H${}_{2}$O at a magnification of 2500× and 50,000×, respectively; (**d**,**e**) as nanofibers produced by electrospinning with Cyclo(L-Trp-L-Tyr) embedded into the biopolymer PCL at a magnification of 5000× and 100,000×. The respective diameter histograms of the nanofibrils (**c**) and the nanofibers (**f**) are represented with dashed curves, indicating lognormal distributions.

**Figure 4 materials-16-04993-f004:**
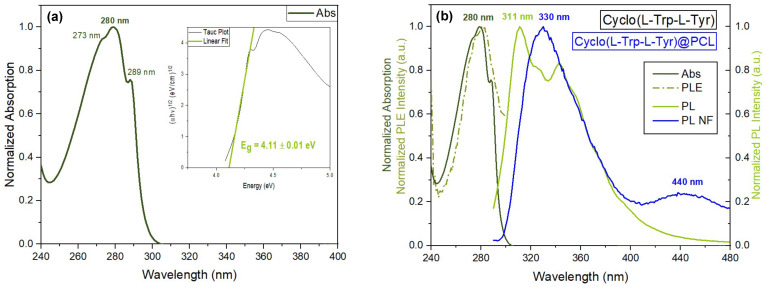
(**a**) Normalized UV–vis absorption spectrum with the bandgap energy obtained from the calculation of the Tauc plot (inset). (**b**) Normalized UV–vis absorption, photoluminescence excitation (PLE), photoluminescence (PL) of Cyclo(L-Trp-L-Tyr) in MeOH and PL spectra of Cyclo(L-Trp-L-Tyr)@PCL nanofibers. (**c**) DRS of Cyclo(L-Trp-L-Tyr)@PCL inset represents the bandgap obtained from the Kubelka–Munk function. (**d**) Confocal microscopy image of the Cyclo(L-Trp-L-Tyr)@PCL nanofiber mat under a laser excitation of 405 nm.

**Figure 5 materials-16-04993-f005:**
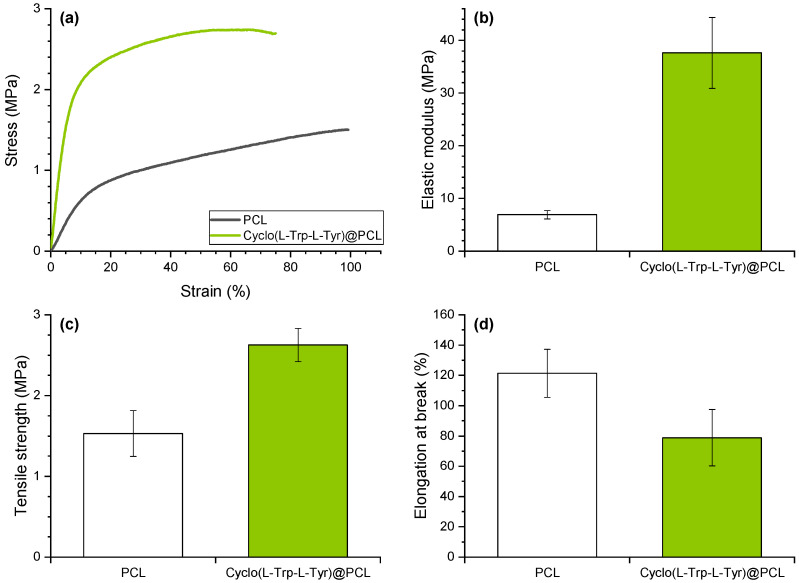
(**a**) Representative tensile curves, (**b**) Elastic modulus (Young’s modulus), (**c**) Tensile strength and (**d**) Elongation at break of the electrospun nanofibers of PCL (grey) and Cyclo(L-Trp-L-Tyr)@PCL (green).

**Figure 6 materials-16-04993-f006:**
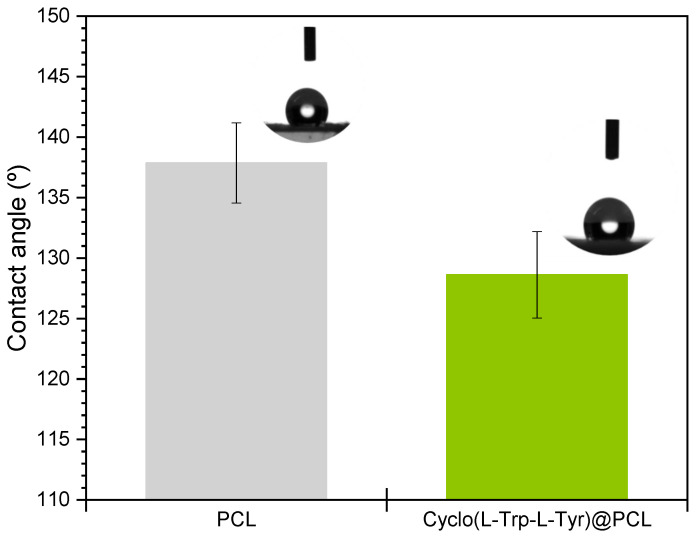
Contact angle of PCL (grey) and Cyclo(L-Trp-L-Tyr)@PCL (green) nanofibers.

**Figure 7 materials-16-04993-f007:**
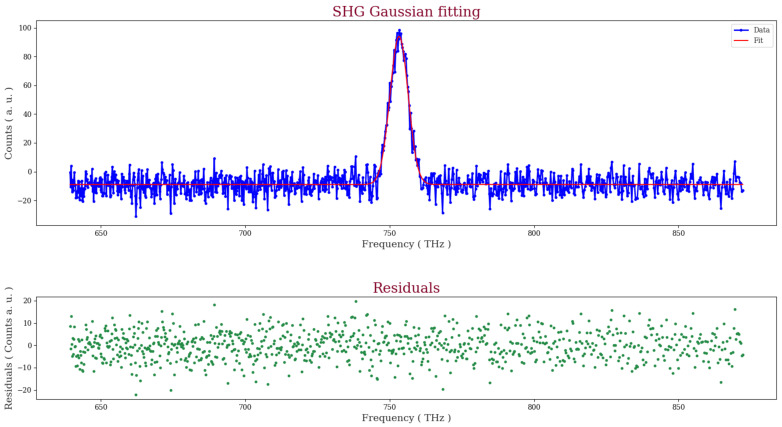
Gaussian fit for the SHG signal from the Cyclo(L-Trp-L-Tyr)@PCL fiber mat.

**Figure 8 materials-16-04993-f008:**
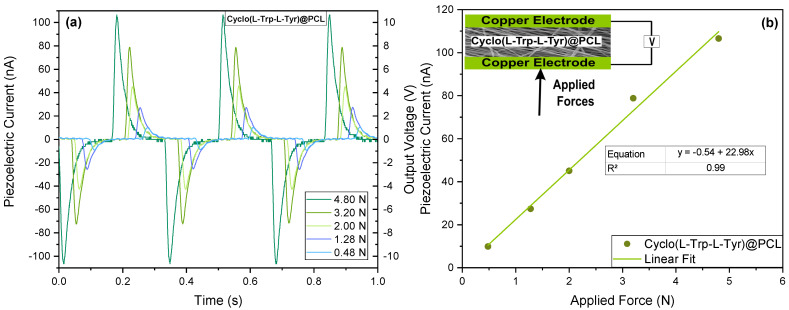
(**a**) Piezoelectric current and output voltage as functions of time; (**b**) maximum piezoelectric current as a function of the different periodical forces applied to Cyclo(L-Trp-L-Tyr)@PCL nanofibers (inset shows the schematic representation of the PENG).

**Table 1 materials-16-04993-t001:** Young’s modulus for different materials.

Sample	Young’s Modulus (MPa)	Ref.
PCL fibers	7	This work
Cyclo(L-Trp-L-Tyr)@PCL fibers	38	This work
Cyclo(L-Trp-L-Trp)@PCL fibers	20	[16]
PVC fibers	4	[33]
MDABCO-NH${}_{4}$I${}_{3}$@PVC fibers	58	[33]
PPU ^1^	9	[35]
PPU/CNC nanocomposite films ^2^	10–150	[35]
PDLLA ^3^	120	[36]
QL4/PDLLA-Polymer composite ^4^	600	[36]

^1^ Peptidic polyurea (PPU). ^2^ Cellulose nanocrystal (CNC). ^3^ Poly(d-,l-lactic acid) (PDLLA). ^4^ 8-amino acid d,l-cyclic peptide (QL4).

**Table 2 materials-16-04993-t002:** Piezoelectric parameters for application in a nanogenerator based on organic compounds.

Piezoelectric Nanogenerator	$d_{\mathrm{eff}}$	$V_{\mathrm{out}}$	Force/Area	Power Density	$g_{\mathrm{eff}}$	Ref.
(PENG)	(pC/N)	(V)	(N/m${}^{2}$)	(μWcm${}^{-2}$)	(Vm/N)	
Cyclo(L-Trp-L-Tyr)@PCL (1:5) (fiber mat)	22	10.5	$7\times 10^{3}$	0.16	1.2	This work
Cyclo(L-Trp-L-Trp)@PLLA (1:5) (fiber mat)	57	11.5	$3\times 10^{3}$	0.18	4.7	[16]
Cyclo(L-Trp-L-Trp)@PCL (1:5) (fiber mat)	30	9.6	$5\times 10^{3}$	0.13	2.6	[16]
Cyclo(FW) (crystal powder)	16 ^1^	1.4	$6\times 10^{5}$	0.003	1.3 ^2^	[11]
Cyclo(GW) (crystal powder)	5.6 ^3^	1.2	$7\times 10^{5}$	0.002	1.6 ^4^	[13]
MDABCO-NH${}_{4}$I${}_{3}$@PVC (1:5) (fiber mat)	175	16.5	$11\times 10^{3}$	0.20	3.6	[33]
Boc-PheTyr@PLLA (fiber mat)	7	24	$4\times 10^{3}$	1.0	0.3	[41]

^1^ Calculated from data available in [11] and assuming a time response of 0.5 s for the nanogenerator; ^2^ Calculated from data available in [11], assuming a dielectric constant of 10; ^3^ Calculated from data in [13]; ^4^ Calculated assuming a dielectric constant of 2.9 from [13].

## Data Availability

Not applicable.

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
