# Peer review of "Nanostructured Electrospun Fibers with Self-Assembled Cyclo-L-Tryptophan-L-Tyrosine Dipeptide as Piezoelectric Materials and Optical Second Harmonic Generators"

_materials, 2023, doi:10.3390/ma16144993_

Round 1

Reviewer 1 Report

The manuscript entitled “Nanostructured Electrospun Fibers with Self-assembled Cyclo-L-Tryptophan-L-Tyrosine Dipeptide as Piezoelectric Materials and Optical Second Harmonic Generators” is in line with the Materials journal. This article is based mainly on original research. The topic is up-to-date and presents new knowledge. The abstract is informative enough. The manuscript has a proper composition and requires only minor revisions before publication, including:

·       The abbreviations should be developed when they are use first time in the text, for example, line 126 (but also others).

·       The link to the website should be given as reference (line 184).

·       Figure 3 should be placed closer to the place when reference is mentioned in the text.

·       Discussion with literature in parts 3.1. and 3.2. is quite generic.

Author Response

Response to Reviewer 1

We thank you very much for your comments and suggestions, which we address below.

“The manuscript entitled “Nanostructured Electrospun Fibers with Self-assembled Cyclo-L-Tryptophan-L-Tyrosine Dipeptide as Piezoelectric Materials and Optical Second Harmonic Generators” is in line with the Materials journal. This article is based mainly on original research. The topic is up-to-date and presents new knowledge. The abstract is informative enough. The manuscript has a proper composition and requires only minor revisions before publication, including:”

  • The abbreviations should be developed when they are use first time in the text, for example, line 126 (but also others).

Author´s response: We have addressed the issue regarding the abbreviations in the manuscript. In the revised document, we have carefully identified and marked all abbreviations, ensuring that they are developed and expanded when they are used for the first time in the text. Thank you for bringing this concern to our attention.

  • The link to the website should be given as reference (line 184).

      Author´s response: Thank you for your valuable suggestion. We have included the following reference in the revised manuscript:

Kortüm, G.; Braun,W.; Herzog, G. Principles and Techniques of Diffuse-Reflectance Spectroscopy. Angewandte Chemie International 571

Edition in English 1963, 2, 333–341. https://doi.org/10.1002/anie.196303331

This reference provides comprehensive information on reflectance spectroscopy, which was relevant to the methodology and analysis employed in our study.

  • Figure 3 should be placed closer to the place when reference is mentioned in the text.

      Author´s response: We have taken your suggestion into consideration, and in the revised version of the manuscript, we have placed Figure 3 closer to the relevant reference in the text as suggested.

  • Discussion with literature in parts 3.1. and 3.2. is quite generic.

Author´s response: We are not sure about the meaning of your question. Nevertheless, we have revised and extended our discussion, incorporating two additional reference to support the statements, as follows:

In our current study, we utilized the dipeptide Cyclo(L-Trp-L-Tyr), taking advantage of the tyrosine OH group's ability to form additional hydrogen bonds with both the tryptophan units and within the tyrosine residues. These hydrogen bonds, facilitated by the tyrosine OH group, are considered stronger than those formed by the tryptophan NH group. This increased strength is attributed to the higher electronegativity of oxygen in the tyrosine OH group compared to nitrogen in the tryptophan NH group, resulting in a greater attraction of shared electrons within the hydrogen bond [27, 28]. Furthermore, the linear arrangement of the hydroxyl (OH) group in tyrosine contributes to enhanced interactions. This is due to the electron-withdrawing nature of the hydroxyl substitution on the phenyl side chain of tyrosine, promoting more efficient π-π interactions involving the side chains [29].

Additionally, it is important to note that tryptophan exhibits a higher hydrophobicity index (1.9) compared to tyrosine (-0.7) [30]. These differences in hydrophobicity can play a role in the self-assembling process. When combined with the enhanced strength of the hydrogen bonds, they facilitate the formation of a more extensive and organized network, thereby promoting the formation of nanotubes.

Reviewer 2 Report

The authors of "Electrospun nanostructured fibers with self-assembly Cyclo-L-Tryptophan-L-Tyrosine Dipeptides as Piezoelectric Materials and Optical Second Harmonic Generators" presents a study on a piezoelectric material essentially a wide bandgap semiconductor (bandgap energy 4.0 eV) and consists of self-assembled nanotubes embedded in a polymer matrix. The paper in this format is not ready for publication. I recommend authors to use the Materials 2023 journal template and to follow it carefully. A few remarks on this version of the paper may contribute to a better future development.

1.    In the introduction, the novelty of this experimental research in relation to the literature should be clearly stated.

2.     The purpose of each experimental investigation (methods) and the connections between them should be specified.

3.      The introduction is excessively long and sometimes appearing to have been added for the purpose of citing another reference.

4.       The methods of investigation are multiple and I believe that all locally available investigations have been done, but for what purpose please set the objective of the research and motivate the investigations in relation to the objective. In its present form it is routine research, still something interesting.

5.       The comparative study is welcome and should be insisted here is a very good part of the article.

6. It is very important to present the experimental setup and the equipment used to measure the piezoelectric properties.

7.       I recommend correlating the piezoelectric performance with the other investigations in order to draw more pertinent conclusions.

8.       The use of a material in real applications implies exceptional performance for it has properties of one kind or another does not automatically qualify it for applications. In this respect I recommend a clear distinction between the case of academic research aimed at knowledge and research aimed at well-objectified applications.

Author Response

Response to Reviewer 2

We thank you very much for your comments and suggestions, which we address below.

“The authors of "Electrospun nanostructured fibers with self-assembly Cyclo-L-Tryptophan-L-Tyrosine Dipeptides as Piezoelectric Materials and Optical Second Harmonic Generators" presents a study on a piezoelectric material essentially a wide bandgap semiconductor (bandgap energy 4.0 eV) and consists of self-assembled nanotubes embedded in a polymer matrix. The paper in this format is not ready for publication. I recommend authors to use the Materials 2023 journal template and to follow it carefully. A few remarks on this version of the paper may contribute to a better future development.”

Author´s response: We would like to clarify that the editors did not mention any specific concerns regarding the format of the paper or the use of the MDPI Overleaf template. Additionally, we want to highlight that the version of the template we used was updated in December 2022, according to the information provided on the MDPI website.

  1. In the introduction, the novelty of this experimental research in relation to the literature should be clearly stated.

Author´s response: Thank you for your suggestion. We have included a new paragraph in the introduction, as follows:

This study represents an innovative contribution in the field, as it is the first time that Cyclo-L-Tryptophan-L-Tyrosine dipeptides have been integrated into a fibrous matrix. We used the electrospinning technique to promote the immediate self-assembling of nanostructures within the fibers. As a result, we achieved a nanofibrous structure with piezoelectric and optical second harmonic generation properties. It is important to emphasize that self-assembling is typically reported in solution, making this approach particularly innovative.

  1. The purpose of each experimental investigation (methods) and the connections between them should be specified.

Author´s response: We appreciate your feedback regarding the need to specify the purpose of each experimental investigation. In the revised version of the manuscript, we have integrated paragraphs, in section 2, that clearly justify the necessity of each characterization method employed in our study. These paragraphs are properly marked and indicate the specific purpose of each experimental investigation, as well as the connections and relevance between them.

  1. The introduction is excessively long and sometimes appearing to have been added for the purpose of citing another reference.

Author´s response: Thank you for your feedback regarding the length of the introduction. We appreciate your input, and we have taken it into consideration in the revised version of the manuscript. We have provided a more concise and focused overview of the study in the introduction, while ensuring that the content remains informative and relevant. Furthermore, we have made efforts to reduce unnecessary citations and improve the clarity and readability of the manuscript.

  1. The methods of investigation are multiple and I believe that all locally available investigations have been done, but for what purpose please set the objective of the research and motivate the investigations in relation to the objective. In its present form it is routine research, still something interesting.

Author´s response: We appreciate and value your comments. The multiple methods of investigation used in our research were aimed at comprehensively characterizing the fibrous matrix and assessing its potential applications and limitations. Through techniques such as SEM and SHG, we gained valuable insights into the material's morphology and optical properties.

The objective of our research was to understand the material's behaviour and explore its potential suitability for integration in piezoelectric and optical second harmonic generation (SHG) devices. We believe that our study goes beyond routine research by providing meaningful insights and opening up possibilities for further exploration.

  1. The comparative study is welcome and should be insisted here is a very good part of the article.

Author´s response: We are not sure about the full meaning of the Reviewer question, in particular to which sections of the manuscript is the Reviewer referring to. However, we have enhanced and expanded specific aspects of the discussion section, along with the inclusion of additional references to substantiate our arguments. These modifications have been clearly indicated in the revised version of the article.

  1. It is very important to present the experimental setup and the equipment used to measure the piezoelectric properties.

Author´s response: We acknowledge the importance of presenting the experimental setup and the equipment used to measure the piezoelectric properties. In the revised version of the manuscript, we have included a comprehensive image in the Supplementary Information that showcases all the equipment utilized in our experiments. This image provides a clear visual representation of the experimental setup, allowing readers to understand the methodology and instrumentation employed.

Additionally, we have provided a detailed description of the experimental setup in the Materials and Methods section (section 2.9.), specifically highlighting the equipment used for measuring the piezoelectric properties.

  1. I recommend correlating the piezoelectric performance with the other investigations in order to draw more pertinent conclusions.

Author´s response: We would like to call the Reviewer attention to Table 1 in the manuscript where several parameters are comparatively displayed for other published organic compounds. We have extended the discussion in this section, as follows:

“Aiming at quantifying the nanofiber mats as a piezoelectric nanogenerator, a figure of merit given by the piezoelectric voltage coefficient geff = deff/(ε´ε0) VmN-1 needs to be calculated.

From reported data the dielectric constant of Cyclo(L- Trp-L-Tyr)@PCL is  ε´~ 2  at low frequencies. Therefore we obtain geff = 1.2 VmN-1 for Cyclo (L-Trp-L-Tyr)@PCL fiber mat. This value is within the same order of magnitude as that calculated for Cyclo (L-Trp-L-Trp)@PCL fiber mat (geff = 2.6 VmN-1).

Finally, the peak power density given by P=(RI^2)/A (μWcm-2), where R=100 MΩ is the load resistance and A (6 cm2) is the electrodes are, delivered by the nanofiber mat is 0.16 μWcm-2, very similar to that power density delivered by Cyclo (L-Trp-L-Trp)@PCL fibers (0.13 μWcm-2).

From Table 1, we arrive to the conclusion that cyclic dipeptides embedded into electrospun fibers display higher piezoelectric properties that those of polycrystalline powder of cyclic dipeptides and linear dipeptides. Therefore, they show great potential to be used in energy harvesting devices through the piezoelectric effect.”

  1. The use of a material in real applications implies exceptional performance for it has properties of one kind or another does not automatically qualify it for applications. In this respect I recommend a clear distinction between the case of academic research aimed at knowledge and research aimed at well-objectified applications.

Author´s response: We have carefully reviewed the article and have taken into account your recommendations. The revised version includes a clear distinction between academic research aimed at knowledge and research focused on well-objectified applications. We have ensured that the manuscript reflects this distinction and provides a more comprehensive understanding of the material's potential applications. The present study is at an academic level and intends to call the attention of researchers, which are oriented to applications development of materials properties.

Reviewer 3 Report

The paper can be accepted after the following corrections:

1. Figure 1 is not suitable for scientific publication. Please re-draw it in the more formal way.

2. Frequency in the figure 3 is misleading. Please clarify.

3. Conclusions about piezoelectric harvesting should be developed and clearly specified in more quantitative way.

Author Response

Response to Reviewer 3

We thank you very much for your comments and suggestions, which we address below.

  1. Figure 1 is not suitable for scientific publication. Please re-draw it in the more formal way.

Author´s response: We understand your concern about the suitability of Figure 1 for scientific publication. As per your request, we have provided a re-drawn version of the figure in a more formal and appropriate manner. Thank you for bringing this to our attention.

  1. Frequency in the figure 3 is misleading. Please clarify.

Author´s response: The label in Figure 3 has been replaced by “Counts”, which will clarify the representation of the data. We appreciate the recommendation.

  1. Conclusions about piezoelectric harvesting should be developed and clearly specified in more quantitative way.

Author´s response: We have carefully reviewed the data presented in the manuscript and will make the necessary revisions to ensure that the conclusions are more quantitative and well-defined. Thank you for your recommendation. In addition to revising the conclusions, we have also made further alterations to the abstract.

The Conclusions are now:

“Biopolymer polycaprolactone (PCL) electrospun fibers with self-assembled cyclic dipeptide L-Tryptophan-L-TyrosiIn nanotubes (Cyclo(L-Trp-L-Tyr)@PCL) form flexible and mechanically resistant fiber mat showing intense blue photoluminescence. The dipeptide nanotubes are wide-bandgap semiconductors with gap energy of 4.0 eV as calculated from the diffuse reflectance spectrum measured in the range 300 nm to 800nm. By embedding the dipeptide into the polymer matrix there is an increase of the Young’s modulus by 445% and the tensile strength by 72% when compared with dipeptide free polymer fibers.

Additionally, the Cyclo(L-Trp-L-Tyr)@PCL fibers demonstrate great potential as optical second harmonic nanogenerators as they present a nonlinear optical effective coefficient of  0.36 pm/V. This value is approximately six times lower than the corresponding coefficient observed in an optical phase-matched inorganic beta-barium borate single crystal. Although the effective coefficient is smaller, for nanophotonic applications, it is much simpler and manageable to work with dipeptide polymer nanofiber mats than with inorganic brittle single crystals which have to be cut with a specific crystallographic orientation.

In order to investigate the feasibility of using this chiral cyclo-dipeptide as a piezoelectric energy harvesting device, periodic forces were applied to the fiber mat, revealing a strong effective piezoelectric coefficient of 22 pCN−1.

Futhermore, the piezoelectric voltage coefficient is geff = 1.2 VmN-1 and the peak power density delivered by the nanofiber mat is 0.16 μWcm-2, very similar to those reported to Cyclo (L-Trp-L-Trp)@PCL fibers, which are  respectively 2.6 VmN-1  and 0.13 μWcm-2.

As a final conclusion, the obtained results provide evidence for the considerable potential of the cyclic-dipeptide L-Tryptophan-L-Tyrosine, when incorporated into electrospun PCL nanofibers, as optical devices and as nanogenerators for energy harvesting technologies.”

Round 2

Reviewer 2 Report

The authors of the revised version of the paper “Electrospun nanostructured fibers with self-assembly Cyclo-L-Tryptophan-L-Tyrosine Dipeptides as Piezoelectric Materials and Optical Second Harmonic Generators” have substantially completed this version with graphic and explanatory content in line with the recommendations in the original version. The work in this version is very well developed and structured in accordance with materials practice.  In its present form, I consider that the paper meets the conditions for publication.

Reviewer 3 Report

The paper was corrected and can be accepted in the present state.